# The MoveStrong program for promoting balance and functional strength training and adequate protein intake in pre-frail older adults: A pilot randomized controlled trial

Isabel B. Rodrigues[1], Ellen Wang[1], Heather Keller[1,2], Lehana Thabane[3], Maureen C. Ashe[4], Sheila Brien[5], Angela M. Cheung[6], Larry Funnell[5], Ravi Jain[5], Desmond Loong[7], Wanrudee Isaranuwatchai[7,8], Jamie Milligan[9], Marina Mourtzakis[1], Alexandra Papaioannou[3,9], Sharon Straus[7,10], Zachary J. Weston[11,12], Lora M. Giangregorio[1,2]*

1 Department of Kinesiology and Health, University of Waterloo, Waterloo, ON, Canada, 2 Schlegel-University of Waterloo Research Institute for Aging, University of Waterloo, Waterloo, ON, Canada, 3 Department of Health Research Methods, Evidence and Impact, McMaster University, Hamilton, ON, Canada, 4 Department of Family Practice, University of British Columbia, Vancouver, BC, Canada, 5 Canadian Osteoporosis Patient Network, Osteoporosis Canada, Toronto, ON, Canada, 6 Department of Medicine, University Health Network, University of Toronto, Toronto, ON, Canada, 7 CLEAR Health Economics, Knowledge Translation Program, Li Ka Shing Knowledge Institute, St. Michael's Hospital, Unity Health Toronto, Toronto, ON, Canada, 8 Institute of Health Policy, Management & Evaluation, University of Toronto, Health Sciences Building, Toronto, ON, Canada, 9 Department of Family Medicine, McMaster University, Hamilton, ON, Canada, 10 Department of Geriatric Medicine, University of Toronto, Toronto, ON, Canada, 11 Waterloo Wellington Local Health Integration Network, Waterloo, ON, Canada, 12 Faculty of Science, Wilfrid Laurier University, Waterloo, ON, Canada

* lora.giangregorio@uwaterloo.ca

**Data Availability Statement:** All relevant data are within the manuscript.

## Abstract

### Background

Balance and functional strength training can improve muscle strength and physical functioning outcomes and decrease the risk of falls in older adults. To maximize the benefits of strength training, adequate protein intake is also important. However, the number of older individuals that consume enough protein or routinely engage in strength training remains low at less than 5% and even lower for activities that challenge balance. Our primary aim was to assess the feasibility of implementing a model (MoveStrong) of service delivery to teach older adults about balance and functional strength training and methods to increase protein intake.

### Methods

This study was a closed cohort stepped wedge randomized controlled trial. We recruited individuals ≥60 years considered pre-frail or frail with at least one chronic condition who were not currently engaging in regular strength training from Northern (rural) and Southern (urban) Ontario sites in Canada. The primary outcome was feasibility of implementation, defined by recruitment, retention, and adherence, and safety (defined by monitoring adverse

**Funding:** LMG: Funding for this project was received from the Canadian Institutes of Health Research catalyst grant: SPOR Innovative Clinical Trials (grant number SCT-162968). The funders had no role in study design, data collection and analysis, decision to publish, or preparation of the manuscript.

**Competing interests:** The authors have declared that no competing interests exist.

events). We also reported participants' and providers' experience with MoveStrong, adaptations to the model based on participant's and provider's experience, and program fidelity.

## Results

We recruited 44 participants to the study and the average adherence rate was 72% with a retention of 71%. The program had a high-fidelity score. One person experienced a fall-related injury during exercise, while two other participants reported pain during certain activities. Five individuals experienced injuries or health problems that were not related to the program. Suggestions for future trials include modifying some exercises, exploring volunteer assistance, increasing the diversity of participants enrolled, and considering a different study design.

## Conclusions

Our pilot trial demonstrates the feasibility of recruitment and adherence for a larger multisite RCT of balance and functional strength training with attention to protein intake in pre-frail and frail older adults.

## Introduction

Balance and strength training can improve muscle strength and physical functioning and disability outcomes, and decrease the risk of falling in older adults with chronic conditions [1–6]. For example, for older adults with sarcopenia, strength training performed between 3 to 18 months, improved muscle mass and strength, and physical performance outcomes such as the chair rise, stair climb, and the 12-minute-walk-test [4, 7, 8]. Progressive strength training performed two to three times per week at a high intensity resulted in moderate to large improvements in gait speed, getting out of a chair, and muscle strength [9]. Furthermore, a Cochrane review reported that balance and functional exercises reduced the rate of falls by 24% in community dwelling older adults [5], and balance and functional exercises in combination with strength training could potentially reduce the rate of falls by more than 30% [5].

The benefits accrued from exercise are evident, but less than 5% of adults 60 years of age and older regularly perform two days a week of balance and strength training [10]. The biggest challenge is not a lack of evidence that balance and functional strength training is beneficial, but the absence of effective and sustainable real-world models for implementation of balance and functional strengthening exercises, especially for older adults with chronic conditions [10, 11]. There are a few, well-designed home and facility-based exercise programs that reduce falls and increase short-term physical activity levels in older adults [12–14]. However, there are few feasible, sustainable, and cost-effective models to deliver balance and functional strength training in a real-world setting [15, 16]. There is also limited evidence on how to effectively implement and sustain these types of exercise interventions in practice, especially for older adults with chronic conditions [15]. To address these challenges, we collaborated with several stakeholders to create MoveStrong–a model of service delivery that provides education and training on performing balance and functional strength training aligned with movements performed during activities of daily living for pre-frail and frail older adults. To maximize the benefits of strength training, we provided nutrition education on consuming enough protein, to support muscle growth and function [17]. Adequate protein intake is a prerequisite to allow net muscle

protein accretion after strength training [17, 18]. Many older adults do not eat enough protein and consume less than 0.66 grams/kg of body weight/day [19]. Food intake in older adults is extremely complex and there are several barriers to consuming enough protein including poor health, poor appetite, changes in food preference, chewing difficulties, etc [20]. A population-based study (n = 2,066 community-dwelling adults 70 to 79 years) showed that those consuming at least 1.2 grams of protein/kg of body weight/day lost lean mass over the three-year follow-up period, but this loss was 40% lower compared to those consuming 0.8 grams of protein/kg of body weight/day [21]. Initiating exercise when protein intake is inadequate may cause weight loss, or limit capacity or strength gains.

The purpose of this study was to conduct a pilot of the MoveStrong program to assess the feasibility of implementation for a larger pragmatic trial. This pilot study assessed the feasibility, fidelity, and the adaptability of a stepped wedge trial evaluating the MoveStrong model in diverse settings across Ontario, Canada. The primary objective was feasibility determined by: 1) evaluating the number of participants recruited to participate; 2) determining retention rates at the end of the study; and 3) calculating adherence rates to the program. We also reported the participants' and providers' experience with MoveStrong, adaptations to the model based on participants' and providers' experiences, and program fidelity. The secondary outcomes (body weight, gait speed, grip strength, fatigue levels, lower limb muscle strength, dynamic balance, health-related quality of life, resource use, and protein intake) were reported in another manuscript [22].

## Methods

We conducted this study in accordance with the extension of the CONSORT 2010 reporting guidelines for stepped-wedge cluster randomized trials (S1 Appendix) [23] and pilot and feasibility trials [24]. We also used the TIDieR (Template for Intervention Description and Replication) checklist to promote full and accurate description of the intervention [25].

### Trial design

The study design was an eight-week pilot, assessor-blinded, multisite, closed cohort stepped wedge randomized controlled trial (RCT). In a stepped wedge design each site was exposed to the intervention but not at the same point in time. Before the program begins, all sites were randomized to start at different time points, each three weeks apart. At regular three-week intervals (the "steps"), one site crosses from the control group to the intervention group [26]. This process continues until all sites have been exposed the MoveStrong program. We selected the stepped wedge design because it provides the advantage that all participants will eventually receive the intervention [26] and this design is used as an alternative to traditional parallel clusters [23]. This design also allowed us to determine the feasibility of using a stepped wedge design for a larger pragmatic trial.

### Study setting

We evaluated the program in areas that typically represent real-world practice and we selected three distinct settings: retirement/assisted living homes, community centers, and a family health team in four cities across Ontario. One northern and three southern Ontario sites were chosen to ensure diversity in city population, structure, and health service. We hired exercise physiologists with at least one-year of experience delivering exercise to older adults so that we could assess the feasibility of real-world implementation rather than have it delivered in a research setting by researchers. We also hired two registered dietitians with the Dietitians of Canada: one dietitian at the northern Ontario site and the other at the southern Ontario sites.

The MoveStrong program was implemented and delivered at a kinesiologist-led clinic partnered with Arbour Trails (retirement/assisted living home and independent living, Guelph, site 1), the City of Lakes Family Health Team (Sudbury, site 2), the Village of Winston Park (retirement/assisted living home and independent living, Kitchener, site 3), and two of the YMCA's of Cambridge and Kitchener-Waterloo (each YMCA is part of one site, site 4). The Sudbury site is located in northern Ontario, while the other three sites are in southern Ontario. The Sudbury site is in rural Ontario, while the other three sites are in urban centers. There are differences between urban and rural populations in terms of health seeking behaviours, health status, and health service use, cost, and outcomes. In general, rural residents have access to a smaller number of health services and providers than urban residents [27].

## Participants

We included participants if they: 1) spoke English or attended with a translator; 2) were ≥ 60 years of age; 3) had a FRAIL Scale score ≥1 (i.e., a score of 0 is robust, a score 1 or 2, pre-frail, and a score of 3 to 5, frail) [28]; and 4) had at least one of the following chronic conditions diagnosed by a physician: diabetes, obesity, cancer (other than minor skin cancer), chronic lung disease, cardiovascular disease, congestive heart failure, hypertension, osteoporosis, arthritis, stroke, or kidney disease. We encouraged participants to attend with a caregiver/friend for social or physical support, and the caregiver/friend was given the opportunity to complete the screening and assessment process to determine if they were eligible to enroll in the study. We excluded individuals who: 1) were currently doing a similar resistance exercise ≥2x/week; 2) were receiving palliative care; 3) could not perform basic activities of daily living; 4) had severe cognitive impairment (e.g., unable to follow two-step commands or could not explain the research study to the research assistant); 5) were travelling >1 week during the MoveStrong program; or 6) had absolute exercise contraindications. We determined absolute exercise contraindications to exercise using the American College of Sports Medicine guidelines [29].

## Recruitment and randomization

We recruited participants from local primary care practices, retirement/assisted living homes, and via advertisement in the local community (e.g., physiotherapy clinics, libraries, and churches) using face-to-face techniques, traditional and social media (Facebook and Twitter), posters, flyers, and brochures. We set up recruitment booths at the two retirement/assisted living home sites. Due to the delay between recruitment and randomization, we decided a priori that participants that dropped out prior to randomization could be replaced up until the intervention started. A biostatistician, independent of the study, created a computer-generated randomization sequence at St. Joseph's Healthcare in Hamilton to randomize sites to start the program at one of four start times, each three weeks apart. A co-investigator at the University of British Columbia kept the randomization sequence concealed and communicated the sequence to all sites after randomization. Each site was assigned to receive the intervention at either week 19, 22, 25, or 28; participants that received the intervention at weeks 22, 25, and 28 were asked to continue their usual activities until the start of the program.

## Intervention

**Exercise program.** The MoveStrong exercise program includes functional strength training movements for older adults of varying abilities, using minimal equipment. Each exercise was informed by the GLA:D program for arthritis [30], BoneFit[TM] [31], and meta-analyses on resistance exercise and fall prevention for older adults [2, 6, 9, 32, 33]. We sought input from representatives from the YMCAs of Cambridge and Kitchener-Waterloo, Community Support

Connections, and Osteoporosis Canada, as well as patient advocates. The exercises are aligned with functional movements to promote personal relevance such as lunging/stepping, reaching, squatting, pulling, lifting and carrying, and pushing (see Table 1 for the TIDieR checklist). Each participant received a one-to-one session with an exercise physiologist (not blinded to site allocation) who selected a starting level and variations for each functional movement, intensity, and the number of repetitions and sets. Then, participants attended an exercise physiologist-led group exercise workshop (1 exercise physiologist to ≤ 6 participants ratio) twice a week for 8-weeks. The exercise program started with a warm-up (5 minutes), followed by the exercise program (50 minutes) and a cool-down (5 minutes). During the first two weeks, the focus was on form rather than on intensity. Exercise difficulty, resistance used, or volume (up to 3 sets, up to 8 reps) was progressed over time, with a target intensity of < 8 repetitions maximum. We did not formally assess 1-repetition maximum. The exercise programs were implemented in real world settings by exercise physiologists, and they were provided a manual and training on how to implement the program. Each exercise physiologist was advised to use informal assessments of multiple repetitions maximum and a repetition in reserve strategy to guide exercise selection and progression. They were instructed to increase the difficulty of the movement if participants could perform more than 8 repetitions. During the cool-down, the exercise physiologist led a 5-minute group discussion where participants discussed when and where to practice the exercise(s) at home or in a setting of choice. Each site received a standardized toolkit with materials for participant workbooks and a trainer manual. The manual provided guidance on how to deliver the workshop, select and progress exercises, adapt exercises for common impairments, cueing tips, and discussion topics. The research team met with each site for one to two-hours to demonstrate how to deliver the MoveStrong program and to review the manual. Participant workbooks were assembled to include pictures and instructions of each exercise so the participants could practice and exercise at home or at another location. Participants received their workbooks during the one-on-one session with the exercise physiologist.

**Nutrition education.** The nutrition intervention included two components: 1) a nutrition education booklet, and 2) two dietitian-led one-hour group seminars to answer questions and discuss topics related to protein intake. The dietitians were not blinded to allocation. The booklet and seminars reviewed the cost of preparing high protein foods, a guide on how and why to spread protein intake through the day, how much protein was in the participant's usual diet, low-cost options to add protein to meals, easy-to-consume protein-rich snacks with minimal preparation, high quality protein supplements (e.g., rapidly digested, high leucine like whey), and how to prioritize high-protein choices in retirement/assisted living home menus or restaurants. During each seminar, the dietitian provided samples of protein-rich snacks. Seminars were held during weeks two and five to allow time to review material, revisit topics, and address questions. We promoted a protein intake of 1.2 grams of protein per kilogram of body weight per day or 20–30 grams of protein per meal [17, 34].

## Outcomes

Outcome assessors were blinded to allocation. There was one outcome assessor at the southern Ontario sites and two outcome assessors at the northern Ontario site. There were four assessments total; all baseline assessments were completed prior to randomization and an additional three assessments were conducted each six weeks apart. The last evaluation was considered as a follow-up assessment.

**Recruitment, retention, and adherence.** The primary outcomes were feasibility of implementation, defined by recruitment (number of participants recruited prior to randomization),

**Table 1. Template for intervention description and replication (TIDieR) checklist.**

| Item Category | Description |
|---|---|
| Brief name | MoveSTroNg: A Model for delivering Strength Training and Nutrition education for older adults in Canadian communities. |
| Why | The benefits accrued from balance and functional strength training in older adults is evident. However, the number of older individuals that routinely engage in strength training remains low at less than 5% and even lower for activities that challenge balance. Novel concepts and models with the potential for large scale implementation and long-term adherence to balance and function strength training are urgently needed for frail older adults. |
| What: *Materials* | The MoveStrong program includes an exercise and a nutrition component. The exercise component is an exercise physiologist led balance and functional strength training program aligned with functional movements to promote personal relevance. Participants are provided an exercise booklet to track their goals and exercises. The nutrition component includes two dietitian-led interactive group seminars to promote strategies to increase protein intake supported by a nutrition booklet. |
| | 1) Participant's exercise booklet: A guide containing a series of pictures and written instructions on how to perform each movement with proper form and technique. There is also a goal setting and planning worksheet. |
| | 2) Participant's nutrition booklets: A guide with tips and recipes complimented by pictures and visual cues on methods to increase protein intake throughout the day. |
| | 3) Instructor's manual: A manual containing information on how to run the exercise programs (e.g., equipment and set-up, how to select and teach each exercise, safety, warm-up and cool down, etc.), cueing tips, and motivational interviewing strategies. |
| | 4) Study manual: A manual containing information about the program timeline, research forms, physical assessment forms, and adverse event reporting forms. |
| | 5) Equipment: All sites received the following equipment: Therabands (3 levels), two sets of Kettlebells (5, 10 and 15 lbs), and step-ups with modifiable levels. |
| What: *Procedures* | The exercise physiologist reviewed each participant's medical history and met with each participant one-on-one prior to the start of the group sessions. The participant and the Kinesiologist selected one of four starting levels for each movement. There were seven functional movements (lunging/stepping, impacting, reaching, squatting, pulling, lifting and carrying, and pushing), and each movement was progressed as necessary. |
| Who: *Provided* | Exercise sessions were delivered by an exercise physiologist with at least one-year of experience working with older adults. The nutrition sessions were offered by an experienced dietitian. |
| How | The exercise program was delivered face-to-face for eight-week in a group setting with one exercise physiologist to five or six participants. Two nutrition sessions were delivered in a group setting with one dietitian to ten participants. |
| Where | We selected sites that represent real-world practice. There were four locations where the program was implemented: 1) Kinnect to Wellness (Sudbury, rural Northern Ontario site); 2) Arbour Trails (retirement/assisted living home and independent living, Guelph, Southern Ontario site); 3) Village of Winston Park (retirement/assisted living home and independent living, Kitchener, Southern Ontario site); and 4) YMCAs of Cambridge and Kitchener (Southern Ontario site). |
| When and how much | Frequency/Duration: 2x/week for 8 weeks, 60 to 90 minutes/session. |
| | Intensity: 2–3 sets of 3–8 repetitions of each exercise with time under tension per repetition of 4:0:2 seconds for eccentric:isometric:concentric phases. |
| Tailoring | Participants began exercising at lower intensity; the focus was on form and technique rather than on effort for the first few weeks. The exercise physiologists increased participants intensity over the first five weeks until participants were at 3 sets of 8 repetitions of each exercise. |
| Modifications | We recruited volunteers to assist the exercise physiologists at the Northern Ontario site and at the retirement/assisted living homes located in the Southern Ontario sites. |

(*Continued*)

**Table 1.** (Continued)

| Item Category | Description |
|---|---|
| How well: *Planned* | A third party, who is not involved in collecting outcome data, assessed if the intervention was delivered and performed as it was intended using a Fidelity Checklist. The average fidelity score was 1.74 out of two. Most exercise physiologists arrived on time, delivered the program the way it was intended, prescribed the correct frequency for each movement, and provided positive reinforcement to encourage participants. Areas where fidelity was an issue were progression of intensity, and completion of the post-exercise discussion regarding doing exercises at home due to a lack of time. |
| How well: *Actual* | The average attendance rate to the MoveStrong program was 72%. The Arbour Trails site had an average adherence rate of 73%, Kinnect to Wellness, 66%, Winston Park, 73%, and the YMCA, 77%. Approximately 62% of individuals attended at least 70% of the exercise sessions (16 sessions total). |

retention (number of participants retained during the follow-up period), and adherence (percentage of exercise sessions completed). The outcome assessors collected the data for recruitment and retention, while the exercise physiologist tracked adherence rates to ensure the outcome assessors were blinded to allocation. Our criteria for success were to recruit 10 participants at each of the four sites or 40 participants total with a retention of 90% at the end of the study, and an adherence of ≥ 70% [35, 36]. We selected a recruitment rate of ten participants at each site (four sites total) because of the proposed class ratio of one instructor to five participants. Recruiting ten participants allowed us to observe the feasibility to deliver two nutrition sessions and two groups of exercise sessions at each site. We allowed sites to over-recruit by up to two participants.

**Participant and provider experience, adaptations, and fidelity.** We conducted exit interviews with participants and the exercise physiologists to understand their experiences. Each interview was audio-recorded, transcribed non verbatim, and analysed using content analysis [37]. In addition, we provided the exercise physiologists and outcome assessors with a spreadsheet to record deviations from the exercise manual and to report any challenges and successes with MoveStrong to inform a future trial. To determine fidelity, a member of the research team with a background in exercise science observed a randomly selected exercise session at each site using a three-point fidelity checklist designed by our research team; a score of 0 indicates the goal was not introduced, a score of 1, the goal was partially achieved, and 2, the goal was fully achieved.

**Adverse events.** Adverse events are unfavourable or unintended occurrence in the health or well-being of a research participant; these events may or may not be related to the intervention [38]. We reported two types of adverse events: 1) serious adverse events defined by Health Canada as "events that result in death, hospitalization, or disability", or 2) minor adverse events. We classified each adverse event as either "not related", "related", or "possibly related" to the intervention. A "not related" category was applied if the participant experienced an adverse event that was not related to the intervention, while a "related" category was utilized if the participant experienced the adverse event that was related to the intervention [39]. A "possibly related" category means there was a reasonable possibility that the event, experience, or outcome may have been caused by the intervention or procedures involved in the trial [39].

## Data safety monitoring committee

The committee comprised of a physiotherapist, a physician, and a biostatistician who reviewed adverse events after three sites completed the program and provided guidance for a future trial. There were no interim analyses and there were no stopping guidelines for the pilot trial.

## Statistical analyses

The analyses and reporting of this trial follow the CONSORT extension to pilot trials [24]. Demographic data, fidelity scores and recruitment, retention and adherence data were reported using means and standard deviations or as a count and percentage. Estimates for feasibility outcomes are reported as percent (95% confidence interval [CI]). Descriptive analyses were performed using SPSS Statistics version 27 (Armonk, New York, USA). We analyzed the exit interviews in NVivo version 12 (QSR International Pty Ltd, 2019) to identify participants' and providers' experiences and suggested adaptations to the MoveStrong program. Adverse events were reported using narrative description.

# Results

Within three months, we screened 75 individuals for eligibility and enrolled 44 participants (Table 2, Fig 1). Within one-week of randomization but prior to the start of the intervention, five individuals dropped out. We enrolled an additional four participants prior to the start of the intervention. Thirty-nine participants started the intervention. After we collected all baseline assessments, each site was randomized to start the intervention at one of four time points between October 2019 to January 2020 (Fig 2). One participant attended with a caregiver who was not enrolled in the program.

## Recruitment, retention, and adherence

Our pilot study took place during October 2019 to March 2020. All participants completed the exercise and nutrition sessions before the COVID-19 pandemic was declared in Canada; however, as a result of the pandemic, we were not able to complete follow-up visits for all participants. Criterion for success for recruitment were met. Of the 75 individuals that demonstrated interest in the program, the main driver for ineligibility was not classifying as pre-frail or frail (94%) or having a health condition that precluded exercise (i.e., uncontrolled asthma or reoccurring acute respiratory infections) (6%). At the southern Ontario sites, the screening to recruitment ratio was 3 potential participants to 1 enrolled participant; however, at the retirement/assisted living homes the screening to recruitment ratio was 6:1. Five individuals withdrew a few days after randomization, and, since the program had not started, we recruited an additional four participants. Reasons for withdrawing after randomization included worsening medical conditions (Kinnect to Wellness, n = 2; Arbour Trail, n = 1; Winston Park, n = 1) or lacking the time to exercise (Kinnect to Wellness, n = 1); three of these five individuals were men with a FRAIL score of 3 and lived alone in a retirement/assisted living home. Criterion for success related to retention were not met; a total of thirteen participants left the study before completion of the final data collection and retention was 70.5% (95% CI 56.4% to 84.5%). Thirty-one of 44 participants completed the study and their follow-up assessment. During the study, two individuals, each from different sites, left due to an injury unrelated to the program. From another site, one individual withdrew due to a minor adverse event possibly related to the intervention, while another participant withdrew because the exercises were too difficult. At follow-up, we lost four participants due to the COVID-19 pandemic (Arbour Trail, n = 1; YMCA, n = 2; Kinnect to Wellness, n = 1). Criterion for success related to adherence to the MoveStrong program were met, with an average adherence rate of 72%, 95% CI 62.7% to 81.6%, 39 participants (Arbour Trails 73%, 8 participants; Kinnect to Wellness 66%, 12 participants; Winston Park 73%, 8 participants, YMCA 77%, 11 participants). Approximately 62% of individuals (95% CI 45.5% to 77.5%) attended at least 70% of the exercise sessions (16 sessions total).

**Table 2. Demographic and health status of trial participants at baseline (n = 44).**

| Characteristics at Baseline | Arbour Trails (n = 9) | Kinnect to Wellness (n = 15) | Village of Winston Park (n = 9) | YMCA (n = 11) |
|---|---|---|---|---|
| | Mean (SD) | Mean (SD) | Mean (SD) | Mean (SD) |
| Age | 78 (11.50) | 81 (5.39) | 84 (8.80) | 72 (7.71) |
| Height (cm) | 161 (10.89), n = 7 | 156 (26.18) | 160 (7.63), n = 7 | 161 (7.71) |
| Body weight (kg) | 72 (19.17), n = 7 | 73 (12.44) | 65 (7.64), n = 8 | 67 (12.80) |
| **Characteristics at Baseline** | **n (%)** | **n (%)** | **n (%)** | **n (%)** |
| Sex (Female) | 7 (78) | 10 (67) | 7 (78) | 10 (91) |
| Ethnicity: | | | | |
| White | 8 (89) | 15 (100) | 8 (89) | 9 (82) |
| South Asian | 0 (0) | 0 (0) | 1 (11) | 2 (18) |
| Middle Eastern | 1 (11) | 0 (0) | 0 (0) | 0 (0) |
| Marital Status: | | | | |
| Married | 2 (22) | 7 (47) | 4 (44) | 7 (64) |
| Widowed | 4 (44) | 6 (40) | 5 (56) | 2 (18) |
| Single/Separated/Divorced | 3 (33) | 2 (13) | 0 (0) | 2 (18) |
| Highest level of education: | | | | |
| Middle school | 0 (0) | 5 (33) | 0 (0) | 1 (9) |
| High School | 0 (0) | 8 (53) | 4 (44) | 3 (27) |
| Higher education (college or university) | 9 (100) | 2 (13) | 5 (56) | 7 (64) |
| Employment: | | | | |
| Retired (not working) | 6 (67) | 15 (100) | 9 (100) | 11 (100) |
| Medical leave | 2 (22) | 0 (0) | 0 (0) | 0 (0) |
| Part-time (<40 hours/week) | 1 (11) | 0 (0) | 0 (0) | 0 (0) |
| Annual income (in Canadian Dollars): | | | | |
| <40,000 | 3 (33) | 7 (47) | 3 (33) | 4 (36) |
| 40,000 to 60,000 | 1 (11) | 5 (33) | 0 (0) | 3 (27) |
| >60,000 | 3 (33) | 0 (0) | 2 (22) | 0 (0) |
| Prefer not to say | 2 (22) | 3 (20) | 4 (44) | 4 (36) |
| Place of residence: | | | | |
| Lives in a retirement home alone | 5 (56) | 1 (7) | 5 (56) | 0 (0) |
| Lives in a retirement home with someone | 0 (0) | 0 (0) | 2 (22) | 0 (0) |
| Lives in the community alone | 2 (22) | 4 (27) | 1 (11) | 4 (36) |
| Lives in the community with someone | 2 (22) | 10 (67) | 1 (11) | 7 (64) |
| Visits from friends and family: | | | | |
| Daily | 3 (33) | 9 (60) | 2 (22) | 1 (9) |
| Weekly | 3 (33) | 5 (33) | 7 (78) | 9 (82) |
| Monthly | 2 (22) | 1 (7) | 0 (0) | 1 (9) |
| Yearly | 1 (11) | 0 (0) | 0 (0) | 0 (0) |
| Use of homecare in the last 6 months | 1 (11) | 1 (7) | 1 (11) | 1 (11) |
| **FRAIL Scale** | **n (%)** | **n (%)** | **n (%)** | **n (%)** |
| How much of the time during the past 4 weeks did you feel tired | 5 (56) | 6 (40) | 5 (56) | 7 (64) |
| Do you have any difficulty walking up 10 steps without resting | 4 (44) | 7 (47) | 4 (44) | 2 (18) |
| Do you have any difficulty walking several hundred yards | 5 (56) | 12 (80) | 8 (89) | 2 (18) |
| Did a doctor ever tell you that you have ≥5 chronic diseases | 3 (33) | 2 (13) | 1 (11) | 0 (0) |
| Weight change >5% in the last year | 3 (33) | 4 (27) | 1 (11) | 4 (36) |
| Average FRAIL score [mean (SD)] | 2.00 (SD 0.50) | 2.07 (0.96) | 2.11 (0.60) | 1.36 (0.67) |
| Two or more components on the FRAIL scale [mean (SD)] | 8 (89) | 10 (67) | 8 (89) | 3 (27) |
| Three or more components on the FRAIL scale [mean (SD)] | 1 (11) | 5 (33) | 2 (22) | 1 (9) |
| **Comorbidities** | **n (%)** | **n (%)** | **n (%)** | **n (%)** |

*(Continued)*

**Table 2.** (Continued)

| Characteristics at Baseline | Arbour Trails (n = 9) | Kinnect to Wellness (n = 15) | Village of Winston Park (n = 9) | YMCA (n = 11) |
|---|---|---|---|---|
| | Mean (SD) | Mean (SD) | Mean (SD) | Mean (SD) |
| Cardiovascular diseases | 4 (44) | 6 (40) | 5 (56) | 2 (18) |
| Hypertension | 8 (89) | 11 (73) | 6 (67) | 4 (36) |
| Respiratory illnesses | 3 (33) | 5 (33) | 2 (22) | 1 (9) |
| Bone disease (Osteoporosis) | 4 (44) | 8 (53) | 5 (56) | 6 (55) |
| Joint disease | 5 (56) | 15 (100) | 6 (67) | 5 (45) |
| Type II Diabetes | 3 (33) | 6 (40) | 2 (22) | 4 (36) |
| Low back pain | 5 (56) | 13 (87) | 4 (44) | 5 (45) |
| **Falls and Fractures** | **n (%)** | **n (%)** | **n (%)** | **n (%)** |
| Number of individuals who fell in the last 6 months | 1 (11) | 4 (27) | 1 (11) | 0 (0) |
| Number of individuals who sustained a fragility fracture in the last 6 months | 0 (0) | 0 (0) | 0 (0) | 0 (0) |
| **Assistive Devices** | **n (%)** | **n (%)** | **n (%)** | **n (%)** |
| Use a walker for mobility | 2 (22) | 0 (0) | 1 (11) | 1 (9) |
| Use a wheelchair for mobility | 1 (11) | 0 (0) | 0 (0) | 0 (0) |

## Adverse events

There were two minor adverse events possibly related to the intervention and one serious adverse event related to the intervention. One participant reported groin strain while exercising but was subsequently diagnosed with hip osteoarthritis. After one-week of rest, this individual returned with a modified exercise program. The second participant had a history of right Achilles tendinitis and complained of ankle pain during the "heel drop" (i.e., impact) movement. Although all lower body exercises were ceased, after one week they withdrew from the study. During the "step-up" movement, one participant sustained an inferior pubic ramus fracture after a fall while stepping down from a one-inch riser. The participant had a history of a femoral neck fracture and possibly lost their balance or experienced muscle weakness while stepping down on the fractured side. The participant had no known mobility or cognitive impairment and did not use an assistive aid; they also had good balance at baseline. Although the exercise physiologist was present and caught them so their whole body did not hit the floor, the exercise physiologist did not catch them in time to completely prevent the fall. The participant did not withdraw from the study but discontinued the exercise program.

There were three minor and two serious adverse events not related to the intervention. One participant slipped in the living room and fractured the metatarsal bones of their left foot. Another participant fell while attempting to sit on an unlocked walker and sustained a right inferior and superior pubic ramus fracture; this participant withdrew from the study. Two participants reported to the emergency room: one with high blood pressure and the other after experiencing a transient ischemic attack. The last participant was at home when they experienced a seizure due to unknown causes and was admitted to the hospital for observation. The participants that experienced the pubic ramus fractures and the seizure were categorized as serious adverse events as a result of being hospitalized.

## Participant and provider experiences: Successes

We interviewed 31 participants and six exercise physiologists. There were three main reasons participants chose to enroll in the program: 1) they had a medical condition that affected their

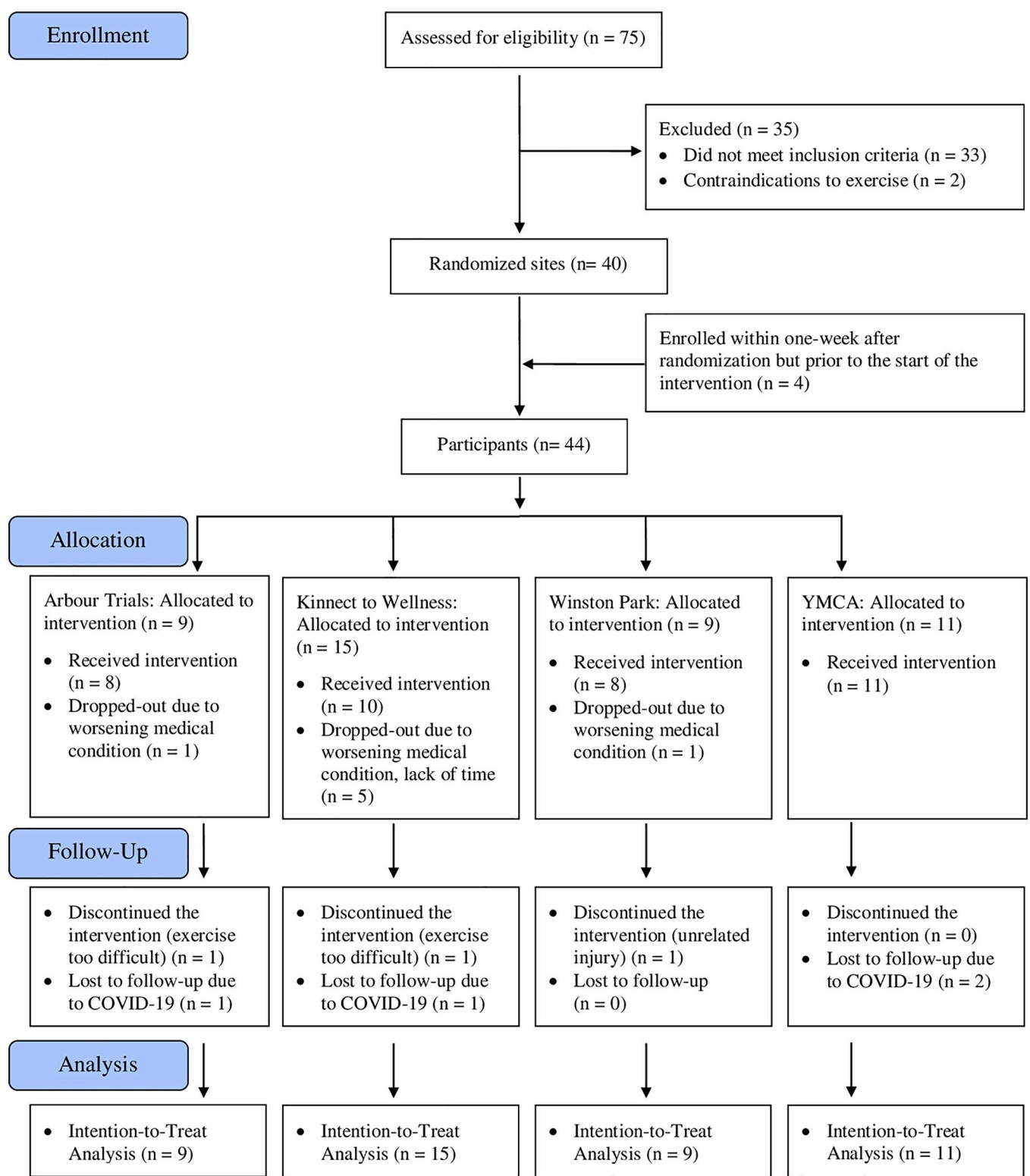

**Fig 1. CONSORT participant flow diagram.**

| Time in weeks | Study start-up, training | Pre-rollout period | 17 to 18 | Rollout, Exposure, and Maintenance | | | | | | | | | | | | | | | | | | Data cleaning, analysis |
|---|---|---|---|---|---|---|---|---|---|---|---|---|---|---|---|---|---|---|---|---|---|---|---|
| | 1 to 8 | 9 to 16 | 17 to 18 | 19 | 20 | 21 | 22 | 23 | 24 | 25 | 26 | 27 | 28 | 29 | 30 | 31 | 32 | 33 | 34 | 35 | 36 | 37 to 52 |
| Site 1 | | | X | | | | | | X | | | | | | X | | | | | | X | |
| Site 2 | | | X | | | | | | X | | | | | | X | | | | | | X | |
| Site 3 | | | X | | | | | X | | | | | | | X | | | | | | X | |
| Site 4 | | | X | | | | | | X | | | | | | X | | | | | | X | |

Training of staff, orientation to study and program, and materials
Pre-rollout period: Study sites initiate advertisement and recruitment
Rollout period: Intervention is rolled out every three-weeks
Exposure period: Participants participate in the MoveStrong program
Maintenance period: MoveStrong program is complete. Participants are encouraged to sustain behaviour change
X     Study visits: Repeated measurements from individuals at fixed time points to allow assessment of period effects

**Fig 2. Timeline for the MoveStrong study.**

daily activities; 2) they were encouraged to join with a friend or family member; or 3) they were encouraged to join by a healthcare professional. Participants reported several benefits from the exercise program including improved posture, strength, balance, self-esteem/confidence, and enhanced social interactions with other participants in the program. Most participants found the nutrition sessions helpful, and most individuals were not aware that they lacked protein in their diets. The term "relearning" was a reoccurring concept during the interview, where participants understood the importance of protein but hearing and discussing the concept again was helpful.

All six exercise physiologists enjoyed delivering the program because of its focus on evidence and small class size. The small class size enabled the instructor to have a more detailed conversation with the participants about their goals. When asked how they implemented the exercise program, the exercise physiologists said they delivered the program in one of two ways: 1) a circuit, where exercises alternated between upper or lower body movements; or 2) a buddy system, where participants with similar levels of intensity were paired. Overall, the exercise physiologists enjoyed the MoveStrong program, and they found the one-on-one sessions provided insight on the participants' needs and goals, which helped guide how the exercise physiologists tailored the program. Most participants and instructors enjoyed the MoveStrong program and stated they would like to see a similar program offered in their retirement/assisted living home or the community center.

### Participant and provider experiences: Challenges

Several participants at the retirement/assisted living homes and the Kinnect to Wellness site stated they felt "weaker" and "limper" after an exercise class, while those enrolled at the YMCA site found the exercises were repetitive after a few weeks. Participants with dentures reported difficulties consuming "hard" food samples (e.g., nuts and seeds). The exercise physiologists at the northern Ontario site and at the retirement/assisted living homes reported the program required a longer time commitment than expected. Certain participants required additional one-on-one support during the exercise session, which took time away from other

participants. The exercise physiologists at the retirement/assisted living homes and the Kinnect to Wellness site reported that some participants found it challenging to initiate the workout on their own and would often wait for assistance; in these instances, a volunteer was recruited at two sites to help other participants. In addition, the exercise physiologists reported participants at the retirement/assisted living homes and at the Kinnect to Wellness site had trouble learning certain movements, with the hip hinge (i.e., during the lift and carry movement) being the most challenging. All the exercise physiologists reported that they spent the first three to five weeks building participants' confidence and focusing on form during each movement, as some participants felt overwhelmed by the number of exercises. The exercise physiologists also suggested that participants with visual impairments and suspected mild cognitive impairment required additional coaching to use the exercise booklet outside of the program.

## Other successes/challenges to implementation

We identified additional challenges during the recruitment and the data collection process using a word document where assessors and exercise physiologists could track successes and challenges. Although we asked all the exercise physiologists to document their experiences with MoveStrong, only one exercise physiologist kept a journal detailing their experiences. Challenges to recruitment observed at the retirement/assisted living home sites were that the participants were not familiar with the research assistant, and some residents felt starting an exercise program at their age was unnecessary. At the YMCA site, there was a response bias with the FRAIL scale, where several participants were enrolled based on the "fatigue" category alone but may not have been pre-frail. We found it challenging to recruit male participants at the southern Ontario sites. There was not enough space or privacy to complete some assessments at the retirement/assisted living homes, some participants were not willing to do the four-square step test for safety reasons or found it difficult, and several participants were not comfortable being weighed. Some participants did not understand certain questionnaire questions. For example, several participants did not associate their children/family members buying groceries and cleaning their homes as assistance with daily activities. Five participants booked vacations between the time of recruitment and the intervention.

## Fidelity

The average fidelity score was 1.74 (0.11); the maximum and highest fidelity score was two. All six exercise physiologists arrived on time, delivered the program the way it was intended, prescribed the correct frequency for each movement, and provided positive reinforcement to encourage participants. Areas where fidelity was an issue were progression of intensity, and completion of the post-exercise discussion regarding doing exercises at home due to a lack of time.

## Discussion

Our feasibility trial provides important lessons that can inform future pragmatic exercise trials in older adults that are pre-frail and frail. We successfully recruited 44 pre-frail or frail older adults in our study, and once recruited, participants exhibited satisfactory adherence. We did not reach our retention goal, in part due to the COVID-19 pandemic. A larger trial may be feasible with some modification to the exercise program. Based on feedback and experience from participants and the exercise physiologists we may need to amend some exercises, consider volunteer assistance, and modify the study design. We suggest removing the step-up exercise for group-based sessions unless it can be done on stairs or steps with a handrail. In addition, a spotter or volunteer assistance may be necessary when participants perform single leg activities

especially participants with a history of hip fracture, even if the participant does not present with balance impairment at baseline. The stepped wedge design may create too large a delay between recruitment and intervention implementation to allow for good retention. The relatively short duration of the program (8-weeks) made it challenging to teach pre-frail and frail individuals strength training and balance exercises and progress intensity in a group setting, so future studies need to consider whether a longer duration or more frequent initial sessions are needed to allocate time to focus on teaching form/technique prior to progressing intensity.

Recruiting frail older adults, male participants, and underrepresented groups to exercise studies can be a complex process. Although we successfully recruited our target number within three months, most of our sample was comprised of pre-frail, older women of white descent. This suggests better strategies to recruit frail individuals of different genders and ethnic backgrounds may be needed. Another challenge was recruiting frail individuals, especially at retirement/assisted living homes, and only 20% of the individuals we recruited were men. One-third of these men dropped out before starting the program and these male participants were frail with mobility impairments and resided alone in a retirement/assisted living home. Our northern Ontario site employed healthcare provider (i.e., nurse) referrals to recruit participants, and we found this to be the most effective method to enroll frail older adults that were both male and female, although all these individuals were of white descent. At our southern Ontario sites, we attempted to recruit diverse groups from places of worship (i.e., temples, synagogues, and mosques) using recruitment posters, though we were not successful. Most researchers have determined there is no single recommended approach to successfully recruit older adults, particularly when attempting to recruit individuals of diverse ethnic backgrounds and genders [40–42]. One study found interpersonal face-to-face approaches such as community talks, physician referrals, and religious leader endorsements were more effective at recruiting Hispanic and African American women than mass mailing or media techniques [43]. A systematic review reported social marketing (i.e., the use of marketing to design and implement programs to promote socially beneficial behavior change), health provider referrals, and referrals from friends, family or other participants in the same study may be the most effective at recruiting diverse groups, although the heterogeneity between the studies was high and may not be generalizable [44]. More resources are needed to recruit and enroll underrepresented groups in exercise research because the benefits may be extended to all populations.

Retention was poor considering that data collection abruptly ended with lockdown due to the COVID19 pandemic. We also did not meet our criteria for success even after over-recruiting participants to account for some loss to follow-up. We lost five participants within one week of randomization; these participants were enrolled one and a half months prior to randomization and during this time a loss of interest or health issues caused them to drop out. We implemented trust-building communication strategies (i.e., through one-on-one sessions with the instructor) and expressed gratitude (e.g., sending holiday cards) to keep participants connected to the study. We lost an additional four participants, who were all categorized as frail, during the study due to medical reasons or because the exercises were too difficult. Our retention rate of 70% was much lower compared to a systematic review of eight randomized controlled trials on exercise in frail older adults that reported a retention rate of 85% or greater [45]. During the follow-up period, we lost an additional four participants due to the COVID-19 pandemic and if it was not for this pandemic, we may have otherwise had a higher retention rate. There is a need for community exercise instructors to monitor exercise programs carefully and recognize how medical history, medications, and prior injuries or adverse events can influence risk of future events. As a result, it is difficult to determine the relative cost-effectiveness of group exercise classes versus individualized exercise programs especially among individuals at high risk of falls and fractures. Overall, there was a low rate of adverse events, and

this consistent with other studies using similar interventions and populations. Strategies to improve retention in future trials may consider stratifying exercise classes by frailty status (i.e., pre-frail versus frail individuals), having volunteer "spotters", and considering a different study design to avoid delays between recruitment and intervention implementation.

Our adherence rates were similar to those previously reported in other trials on exercise in frail older adults [46–49]. A 2011 systematic review by Theou and colleagues reported adherence to an exercise program in frail older adults ranged between 42% to 100% with a mean adherence rate of 84% [50]. Although the mean exercise adherence in this systematic review was higher than our trial, there were no adverse events reported in most of the included studies [50]. A key challenge in exercise trials is that the people that often enroll in the trial want to exercise, so adherence rates may not be representative of pre-frail or frail individuals who may find these exercises difficult. In addition, many exercise studies in older adults exhibit healthy responder bias, whereas in our study, the number of comorbid conditions and incidence of adverse events during follow-up suggest that our sample was more representative of pre-frail older adults. Theou and colleagues supported the use of "exercise as a safe and feasible intervention for this [frail] population"; however, we found that pre-frail and frail older adults require continuous monitoring throughout the program. We experienced several adverse events related and unrelated to the program, which are likely to affect adherence rates and retention.

We acknowledge some limitations in our study. We mainly recruited pre-frail women of white descent so the results may not be generalizable to men, frail populations, or diverse groups. Despite reminding participants at the start of every assessment not to reveal group allocations a few participants forgot or did not understand and by the third outcome assessment all outcome assessors were unblinded. In addition, some of our retention was affected by the COVID-19 pandemic and may not necessarily reflect normal circumstances. Lastly, we did not conduct an exit interview with the dietitians.

## Conclusion

Our pilot trial demonstrates the feasibility of recruitment and adherence for a larger trial of balance and functional strength training with education on protein intake in pre-frail and frail older adults. Although we did not meet our goal for retention, it was in part affected by the COVID-19 pandemic. Recruiting individuals that were frail, male, and underrepresented groups was a challenge and there was a large learning curve for participants to learn the exercises. Suggestions for future trials include modifying some exercises for pre-frail and frail individuals, considering volunteer assistance, employing recruitment strategies to target men and diverse groups, and considering a different study design to avoid delays between recruitment and starting the intervention.

## Supporting information

**S1 Appendix. CONSORT 2010 checklist.**
(PDF)

**S2 Appendix. MoveStrong protocol.**
(PDF)

## Acknowledgments

The authors would like to acknowledge the assistance of several individuals. We would like to thank all the exercise physiologists, Jessica Bodson, Katarina Bubulj, Katelyn Corke, Bridget

Misener, and Eliza Reid, and the dietitians, Kathy Lepp and Nicole Selman, that helped to deliver the program. We would also like to acknowledge Denise Maki and Tina Treitz for allowing us to use their location, Kinnect to Wellness, in Sudbury, Ontario to deliver the program and to recognize Dave Courtemanche, Sarah Crichton, Nathalie Chisholm, and Meghan Peters at the Family Health Teams in Sudbury for helping with recruiting participants and assessing outcomes. In addition, we would like to acknowledge Jennifer Bucino, Josie d'Avernas, Andrea Grantham, Nathan Honsberger, Michael Lewiecki, Shaen Gingrich, Alex Steinke, Justin B. Wagler, and Cindy Wei for their ongoing operational support.

## DSMB members

The DSMB chair was represented by Dr. Stephanie Kaiser from the Division of Endocrinology and Metabolism at Dalhousie University. The DSMB clinical investigator was Dr. Christine Friedenreich, a senior director of the Cancer Epidemiology and Prevention Research Alberta Health Services, and the DSMB biostatistician was Dr. Eleanor Pullenayegum from Dalla Lana School of Public Health at the University of Toronto.

## Other information

### Registration

This trial was registered in ClinicalTrials.gov under identifier NCT04037436.

### Protocol

The protocol can be found here: https://clinicaltrials.gov/ProvidedDocs/36/NCT04037436/Prot_SAP_ICF_000.pdf

## Author Contributions

**Conceptualization:** Heather Keller, Sheila Brien, Angela M. Cheung, Larry Funnell, Ravi Jain, Wanrudee Isaranuwatchai, Marina Mourtzakis, Alexandra Papaioannou, Sharon Straus, Zachary J. Weston, Lora M. Giangregorio.

**Data curation:** Isabel B. Rodrigues.

**Formal analysis:** Isabel B. Rodrigues, Lehana Thabane, Desmond Loong, Wanrudee Isaranuwatchai, Lora M. Giangregorio.

**Investigation:** Isabel B. Rodrigues, Ellen Wang, Maureen C. Ashe, Lora M. Giangregorio.

**Methodology:** Isabel B. Rodrigues, Heather Keller, Lehana Thabane, Maureen C. Ashe, Sheila Brien, Angela M. Cheung, Larry Funnell, Wanrudee Isaranuwatchai, Jamie Milligan, Marina Mourtzakis, Alexandra Papaioannou, Sharon Straus, Zachary J. Weston, Lora M. Giangregorio.

**Project administration:** Isabel B. Rodrigues, Zachary J. Weston, Lora M. Giangregorio.

**Resources:** Maureen C. Ashe, Zachary J. Weston, Lora M. Giangregorio.

**Software:** Lehana Thabane.

**Supervision:** Heather Keller, Lora M. Giangregorio.

**Validation:** Lora M. Giangregorio.

**Visualization:** Lora M. Giangregorio.

**Writing – original draft:** Isabel B. Rodrigues.

**Writing – review & editing:** Isabel B. Rodrigues, Ellen Wang, Heather Keller, Lehana Thabane, Maureen C. Ashe, Sheila Brien, Angela M. Cheung, Desmond Loong, Wanrudee Isaranuwatchai, Marina Mourtzakis, Alexandra Papaioannou, Sharon Straus, Zachary J. Weston, Lora M. Giangregorio.

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
