## [Decision Letter · Decision Letter 0]

29 Jun 2021

PONE-D-21-14688

The MoveStrong program for promoting balance and functional strength training and adequate protein intake in pre-frail older adults: A pilot randomized controlled trial

PLOS ONE

Dear Dr. Rodrigues,

Thank you for submitting your manuscript to PLOS ONE. After careful consideration, we feel that it has merit but does not fully meet PLOS ONE’s publication criteria as it currently stands. Therefore, we invite you to submit a revised version of the manuscript that addresses the points raised during the review process.

While I find the study interesting, there are major concerns that you should adjust before we accept the assignment. Especially, I agree with reviewer 1 where the term RCT was misplaced. Either clarify your study design (how it is controlled and randomized) or change the study design type. Please pay full attention to all points raised by the reviewers.

Minor points:

Line 80 - reference

Tables - consider using just horizontal lines.

We look forward to receiving your revised manuscript.

Kind regards,

Leonardo A. Peyré-Tartaruga, Ph.D.

Academic Editor

PLOS ONE

Reviewers' comments:

Reviewer's Responses to Questions

**Comments to the Author**

1. Is the manuscript technically sound, and do the data support the conclusions?

Reviewer #1: Partly

Reviewer #2: Yes

Reviewer #3: Yes

Reviewer #4: Yes

2. Has the statistical analysis been performed appropriately and rigorously? 

Reviewer #1: N/A

Reviewer #2: Yes

Reviewer #3: Yes

Reviewer #4: Yes

3. Have the authors made all data underlying the findings in their manuscript fully available?

Reviewer #1: Yes

Reviewer #2: Yes

Reviewer #3: Yes

Reviewer #4: Yes

4. Is the manuscript presented in an intelligible fashion and written in standard English?

Reviewer #1: Yes

Reviewer #2: Yes

Reviewer #3: Yes

Reviewer #4: Yes

5. Review Comments to the Author

Reviewer #1: This is generally a well written paper that does what it is supposed to do, evaluating the feasibility of a trial for a service delivery model to increase protein intake. I will focus on methods and reporting. However, there a few major issues with the study, which I cannot really get my head around.

Major

1) the feasibility study does not include controls, unless i'm mistaken. Then the description of the study is wrong, this is not an RCT study. If i'm mistaken, clarification is needed, since it seems centres all recruit cases.

1) Why is the primary outcome of (future) interest, protein intake, not collected and reported? Assuming there are no controls, that makes sense. However, a key issue in a feasibility study is powering the main study to follow. I don't see how the authors will do that through this feasibility study, with the primary outcome of interest not measured (to obtain baseline values in the two groups and variance).

Minor

1) the paper is very clear but also excessively long. That's not a problem for an open access journal but some focus have helped (no action needed). And for all its length it seems to failing to inform properly on the future trial.

Reviewer #2: The authors performed a closed cohort stepped wedge randomized controlled trial to assess the feasibility of implementing a model (MoveStrong) of service delivery to teach older adults about balance and functional strength training and methods to increase protein intake. There were recruited 44 participants at 4 distinct sites. The intervention consisted of a one-to-one session with an exercise physiologist, followed by 8 weeks (1h/2x/week) group exercise workshop supervised by an exercise physiologist (ratio: 1 exercise physiologist to ≤ 6 participants).

The introduction and justification of the study were well given. The objectives and outcomes were clearly defined. The methods section is rich in information. The results and discussion sections were also well developed. The article is written in plain English.

Line 80 – Please adjust citation.

Line 87-88 – It seems to lack a reference for the sentence about protein intake.

Line 117 – the authors refer that one northern and three southern Ontario sites were chosen to ensure participants diversity. Nevertheless, the epidemiological differences between these Ontario regions may be easily distinguishable to Canadians’ readers, I think that could not come so clear to readers of other countries. Maybe if the authors could provide more information about these regions’ differences, it would help the reading for foreign readers.

Line 145 – there are two slashes instead of one.

Line 199 – I think the more properly unit should be “body mass” instead of “body weight”.

Table 2 – At “Fall and Fractures” and “Assistive Devices” sections, the unit headers are lacking. Also, the specification of “% of people who had…” does not make much sense, since probably the data is presented as n (%). Probably, if the unit header is filled, this would come clearer.

Line 347 – the authors refer that some participants presented visual and/or mild cognitive impairment. Which brings the question of how was these impairments measured or determined? Also, a subgroup analysis of these participants could give some interest findings about the protocol feasibility for these specific populations.

Line 427/297 – It is informed that the protocol presented a low rate of adverse events. In Line 297, however, the authors inform that one participant had an adverse event of lower limb fracture due a fall during an exercise of the protocol. Perhaps if the authors could give more details about this event, the characteristics of the participant (visual or cognitive impairment?). And perhaps if this adverse event does not render a discussion about the physiologist professional/participants ratio.

Figure 1 – I do not know if was document .pdf version, but it was a bit difficult to distinguish the colors of the Rollout and Exposure periods. Maybe if the authors could paint them with more contrasting colors would avoid this confusion.

Reviewer #3: The main purpose of the study was to assess the feasibility of implementing a model (MoveStrong) of service delivery to teach older adults about balance and functional strength training and methods to increase protein intake. The study is interesting and well written. I have only few comments that I believe will improve the manuscript.

Introduction, first sentence:

In order to reinforce the literature on the effects of strength combined to balance training to decrease risk of falls in frail individuals, I suggest to add 2 more references:

1) Multicomponent exercises including muscle power training enhance muscle mass, power output, and functional outcomes in institutionalized frail nonagenarians.

Cadore EL, Casas-Herrero A, Zambom-Ferraresi F, Idoate F, Millor N, Gómez M, Rodriguez-Mañas L, Izquierdo M. Age (Dordr). 2014 Apr;36(2):773-85. doi: 10.1007/s11357-013-9586-z.

2) Effects of different exercise interventions on risk of falls, gait ability, and balance in physically frail older adults: a systematic review.

Cadore EL, Rodríguez-Mañas L, Sinclair A, Izquierdo M. Rejuvenation Res. 2013 Apr;16(2):105-14.

Discussion, limitations paragraph:

Unfortunately, as many projects around the world, authors had their study impaired by the COVID-19 situation, So, I suggest to include as limitations that the results were influenced by this context (if I understood), and not necessarily reflects a normal situation (without pandemic).

Reviewer #4: This current manuscript aimed to assess the feasibility, fidelity and adaptability of the MoveStrong model, a model of service delivery that provides education and training on balance and functional strength training, including nutrition education.

Overall, this is a great model and the “real-life” setting in which the trial was performed, is a great addition to the field of exercise in frail older adults. Additionally, the research design is very robust and the manuscript is well-written. The honest reporting of some of the findings, including the adverse events, is very much appreciated and also a great addition to the field of exercise in frail older adults. I have some minor comments and questions, which I identified in each separate section below.

Title page: On the title page it is stated that the ¶ symbol indicates that these authors contributed equally to this work, however, the symbol is not used in the author list.

Abstract: The abstract reads well and is reflective of the manuscript.

Background: Overall, the background is well written, provides up to date references and clearly outlines the rationale for the study. Some minor comments:

- I think the authors could add some more information in regard to the importance of the addition of the nutrition education

- Please provide a reference for the statement in line 73-76.

- Line 78-80 states {Formatting Citation}, please ensure a correct reference.

- Please provide a reference for the statement in line 87-88.

- The authors state that secondary outcomes were reported in another manuscript in line 96-97, is a reference available?

Methods: The study design is reported in great detail and very clear. I have two questions;

- Although I appreciate that the secondary outcomes are reported in a different manuscript, I would advise to briefly describe all the outcome measures. For instance, in line 175 it is stated that the intensity was targeted at <8 repetitions maximum. However, it is unclear when the RM was measured, which exercises etc.

- Details for the following might be described elsewhere, but I would advise to briefly describe the control group, as the title and abstract state it is a randomized controlled trial.

Results: Clearly described and interesting findings.

Discussion: Well written. The discussion highlights important findings and great insights in regard to the chosen modality of providing the MoveStrong model.

6. PLOS authors have the option to publish the peer review history of their article (what does this mean?). If published, this will include your full peer review and any attached files.

Reviewer #1: No

Reviewer #2: **Yes: **André Ivaniski-Mello

Reviewer #3: No

Reviewer #4: **Yes: **Lara Vlietstra

---

## [Author Response · Author response to Decision Letter 0]

3 Aug 2021

Thank you for reviewing our manuscript "The MoveStrong program for promoting balance and functional strength training and adequate protein intake in pre-frail older adults: A pilot randomized controlled trial" and for providing us with valuable feedback. We also highlight all changes made to the manuscript in red font (starts on page 47 of the PDF).

Reviewer #1: This is generally a well written paper that does what it is supposed to do, evaluating the feasibility of a trial for a service delivery model to increase protein intake. I will focus on methods and reporting. However, there a few major issues with the study, which I cannot really get my head around.

Major

1) the feasibility study does not include controls, unless i'm mistaken. Then the description of the study is wrong, this is not an RCT study. If i'm mistaken, clarification is needed, since it seems centres all recruit cases.

The stepped wedge randomized controlled trial (RCT) design that we have used is a type of RCT. It is used as an alternative to traditional parallel cluster RCTs (1). We opted to use a stepped wedge RCT because it provides the advantage that all participants will eventually receive the intervention. When clusters are relatively similar (homogeneous), parallel cluster RCTs tend to deliver better statistical performance than a stepped wedge cluster RCT; however, if there are substantial cluster differences (e.g., population differences such as rural vs urban populations, community-dwelling vs residential), the stepped wedge design will be more powerful than a parallel design . We were considering a stepped wedge RCT for our future pragmatic trial looking at scaling-up of our intervention, so we assessed feasibility using the same design we were considering for the larger trial.

The design includes an initial period in which no clusters are exposed to the intervention. Subsequently, at regular intervals one cluster is randomized to cross from the control to the intervention group. In our case, sites three and four act as our “control group” until all groups are allocated to the intervention group. In addition, the CONSORT group developed a separate reporting guideline for stepped-wedged cluster RCTs (2). To address your point, we have added the following statement: “We selected the stepped wedge design because it provides the advantage that all participants will eventually receive the intervention (26) and this design is used as an alternative to traditional parallel clusters (23). This design also allowed us to determine the feasibility of using a stepped wedge design for a larger pragmatic trial” (page 5, lines 119 to 123). 

References: 

(1) https://trialsjournal.biomedcentral.com/articles/10.1186/s13063-018-3116-3

(2) https://www.bmj.com/content/363/bmj.k1614.long

2) Why is the primary outcome of (future) interest, protein intake, not collected and reported? Assuming there are no controls, that makes sense. However, a key issue in a feasibility study is powering the main study to follow. I don't see how the authors will do that through this feasibility study, with the primary outcome of interest not measured (to obtain baseline values in the two groups and variance).

Protein intake was measured and is a secondary outcome. We will report the results of our secondary outcomes in another manuscript: “The secondary outcomes (body weight, gait speed, grip strength, fatigue levels, lower limb muscle strength, dynamic balance, health-related quality of life, resource use, and protein intake) were reported in another manuscript (22)”, page 4, line 104 to 107). Our primary outcome was feasibility, defined by recruitment, retention and adherence rates, and the results of the primary research questions are presented in this manuscript. We are reporting feasibility and secondary outcomes in separate manuscripts to have enough space to report all quantitative and qualitative data. The reviewer’s comment that the manuscript is long reflects the need to separate the feasibility data (primary) and exploratory efficacy data (secondary) into two manuscripts, as adding all the secondary outcomes into this manuscript would make it excessively long and unfocused.

It is a common misconception that the primary goal of a pilot or feasibility study should be used to power future studies; power estimates from pilot studies may be biased or imprecise because of small sample sizes and variability in effects (3). While one can use a pilot study to estimate sample size, it should be done very cautiously, and the practice is controversial (4). Some argue that the mean difference used in a sample size calculation should be determined by the minimum clinically important difference, and not using data from a pilot trial (5). In accordance with current research and guidance on pilot and feasibility studies, the primary focus of our trial was feasibility, which is based on process measures, like recruitment rates, fidelity, and compliance. 

References: 

(3) https://bmcmedresmethodol.biomedcentral.com/articles/10.1186/1471-2288-10-1

(4) https://pilotfeasibilitystudies.biomedcentral.com/articles/10.1186/s40814-019-0493-7

(5) https://pilotfeasibilitystudies.biomedcentral.com/articles/10.1186/s40814-019-0493-7

Minor

1) the paper is very clear but also excessively long. That's not a problem for an open access journal but some focus have helped (no action needed). And for all its length it seems to failing to inform properly on the future trial.

Thank you – we have included this statement in the discussion to inform a future trial: “A larger trial may be feasible with some modification to the exercise program. Based on feedback and experience from participants and the exercise physiologists we may need to amend some exercises, consider volunteer assistance, and modify the study design. We suggest removing the step-up exercise for group-based sessions unless it can be done on stairs or steps with a handrail. In addition, a spotter or volunteer assistance may be necessary when participants perform single leg activities especially participants with a history of hip fracture, even if the participant does not present with balance impairment at baseline. The stepped wedge design may create too large a delay between recruitment and intervention implementation to allow for good retention. The relatively short duration of the program (8 weeks) made it challenging to teach pre-frail and frail individuals strength training and balance exercises and progress intensity in a group setting, so future studies need to consider whether a longer duration or more frequent initial sessions are needed to allocate time to focus on teaching form/technique prior to progressing intensity” (page 21 to 22, lines 405 to 417). 

Reviewer #2: The authors performed a closed cohort stepped wedge randomized controlled trial to assess the feasibility of implementing a model (MoveStrong) of service delivery to teach older adults about balance and functional strength training and methods to increase protein intake. There were recruited 44 participants at 4 distinct sites. The intervention consisted of a one-to-one session with an exercise physiologist, followed by 8 weeks (1h/2x/week) group exercise workshop supervised by an exercise physiologist (ratio: 1 exercise physiologist to ≤ 6 participants).

The introduction and justification of the study were well given. The objectives and outcomes were clearly defined. The methods section is rich in information. The results and discussion sections were also well developed. The article is written in plain English.

Line 80 – Please adjust citation.

We have updated this citation – reference 15 and 16 on page 3, line 80.

Line 87-88 – It seems to lack a reference for the sentence about protein intake.

We have included two citations to support the benefits of protein intake and muscle protein accretion after strength training in older adults – references 17 and 18 

on page 4, line 88. 

Line 117 – the authors refer that one northern and three southern Ontario sites were chosen to ensure participants diversity. Nevertheless, the epidemiological differences between these Ontario regions may be easily distinguishable to Canadians’ readers, I think that could not come so clear to readers of other countries. Maybe if the authors could provide more information about these regions’ differences, it would help the reading for foreign readers.

We have included the following statement: “The Sudbury site is in rural Ontario, while the other three sites are in urban centers. There are differences between urban and rural populations in terms of population size, health seeking behaviours, health status, and health service use, cost, and outcomes. In general, rural residents have access to a smaller number of health services and providers than urban residents (27)” (page 6, lines 139 to 143). 

Line 145 – there are two slashes instead of one.

We removed the second slash – page 8, line 151. 

Line 199 – I think the more properly unit should be “body mass” instead of “body weight”.

We understand the term body “mass” should be used over the term “weight” since the unit for “weight” is Newtons; however, several studies related to protein intake are based on body weight. For example, the Estimated Average Requirement (EAR), the Recommended Dietary Allowance (RDA), and the Acceptable Macronutrient Distribution Range (AMDR) are based on consuming protein in grams/kg of body weight/day. Because we are using the EAR, RDA and AMDR to interpret data, we would like to use body weight for consistency. 

Table 2 – At “Fall and Fractures” and “Assistive Devices” sections, the unit headers are lacking. Also, the specification of “% of people who had…” does not make much sense, since probably the data is presented as n (%). Probably, if the unit header is filled, this would come clearer.

We have included the follow unit head in Table 2: “Number of individuals who fell in the last 6 months” and “Number of individuals who sustained a fragility fracture in the last 6 months” – page 16. 

Line 347 – the authors refer that some participants presented visual and/or mild cognitive impairment. Which brings the question of how was these impairments measured or determined? Also, a subgroup analysis of these participants could give some interest findings about the protocol feasibility for these specific populations.

We did not assess for cognitive impairment due to the ethical implications of having someone who is not a health care provider reveal an incidental finding of cognitive impairment and not being able to communicate it or provide treatment options. In interviews, the exercise physiologists referred to suspected cognitive impairment, presumably based on inability to follow simple step-by-step tasks, even after repeated explanations. We have made the following modification in the manuscript to clarify this point: “The exercise physiologists also suggested that participants with visual impairments and suspected mild cognitive impairment required additional coaching to use the exercise booklet outside of the program.” (page 20, line 374 to 376) We cannot do subgroup analyses because we did not formally assess visual or cognitive impairment, and did not plan these subgroup analyses a priori.

Line 427/297 – It is informed that the protocol presented a low rate of adverse events. In Line 297, however, the authors inform that one participant had an adverse event of lower limb fracture due a fall during an exercise of the protocol. Perhaps if the authors could give more details about this event, the characteristics of the participant (visual or cognitive impairment?). And perhaps if this adverse event does not render a discussion about the physiologist professional/participants ratio.

We have revised the manuscript to provide more insight on the event and participant characteristics: “During the “step-up” movement, one participant sustained an inferior pubic ramus fracture after a fall while stepping down from a one-inch riser. The participant had a history of a femoral neck fracture and possibly lost their balance or experienced muscle weakness while stepping down on the fractured side. The participant had no known mobility or cognitive impairment and did not use an assistive aid; they also had good balance at baseline. Although the exercise physiologist was present and caught them so their whole body did not hit the floor, the exercise physiologist did not catch them in time to completely prevent the fall. The participant did not withdraw from the study but discontinued the exercise program.” (page 18 line 320 to 327). We cannot comment on visual impairment as we did not assess vision.

Lowering the ratio of participants to exercise physiologists would not be realistic in terms of cost. It is not easy to predict who will fall and fracture and so we also included the following statement in the discussion to discuss modifications for a larger trial: “We suggest removing the step-up exercise for group-based sessions unless it can be done on stairs or steps with a handrail. In addition, a spotter or volunteer assistance may be necessary when participants perform single leg activities especially participants with a history of hip fracture, even if the participant does not present with balance impairment at baseline” (page 21 lines 408 to 412). 

Figure 1 – I do not know if was document .pdf version, but it was a bit difficult to distinguish the colors of the Rollout and Exposure periods. Maybe if the authors could paint them with more contrasting colors would avoid this confusion.

We have modified this image to use more contrasting colours, but the compression algorithm used by the journal is modifying the quality of the image. We will work with the journal to send them a high-quality image. 

Reviewer #3: The main purpose of the study was to assess the feasibility of implementing a model (MoveStrong) of service delivery to teach older adults about balance and functional strength training and methods to increase protein intake. The study is interesting and well written. I have only few comments that I believe will improve the manuscript.

Introduction, first sentence: In order to reinforce the literature on the effects of strength combined to balance training to decrease risk of falls in frail individuals, I suggest to add 2 more references:

1) Multicomponent exercises including muscle power training enhance muscle mass, power output, and functional outcomes in institutionalized frail nonagenarians.

Cadore EL, Casas-Herrero A, Zambom-Ferraresi F, Idoate F, Millor N, Gómez M, Rodriguez-Mañas L, Izquierdo M. Age (Dordr). 2014 Apr;36(2):773-85. doi: 10.1007/s11357-013-9586-z.

2) Effects of different exercise interventions on risk of falls, gait ability, and balance in physically frail older adults: a systematic review.

Cadore EL, Rodríguez-Mañas L, Sinclair A, Izquierdo M. Rejuvenation Res. 2013 Apr;16(2):105-14.

We have included these two references in the manuscript – references 7 and 8 on page 3, line 66 

Discussion, limitations paragraph: Unfortunately, as many projects around the world, authors had their study impaired by the COVID-19 situation, So, I suggest to include as limitations that the results were influenced by this context (if I understood), and not necessarily reflects a normal situation (without pandemic).

We have included a section in the limitations to address this point: “In addition, some of our retention was affected by the COVID-19 pandemic and may not necessarily reflect normal circumstances” (page 25, line 482 to 483). 

Reviewer #4: This current manuscript aimed to assess the feasibility, fidelity and adaptability of the MoveStrong model, a model of service delivery that provides education and training on balance and functional strength training, including nutrition education.

Overall, this is a great model and the “real-life” setting in which the trial was performed, is a great addition to the field of exercise in frail older adults. Additionally, the research design is very robust and the manuscript is well-written. The honest reporting of some of the findings, including the adverse events, is very much appreciated and also a great addition to the field of exercise in frail older adults. I have some minor comments and questions, which I identified in each separate section below.

Title page: On the title page it is stated that the ¶ symbol indicates that these authors contributed equally to this work, however, the symbol is not used in the author list.

The instruction from PLOS ONE suggest using the ¶ symbol on the title page but not in the author list. 

Abstract: The abstract reads well and is reflective of the manuscript.

Background: Overall, the background is well written, provides up to date references and clearly outlines the rationale for the study. Some minor comments:

- I think the authors could add some more information in regard to the importance of the addition of the nutrition education

We have included the following paragraph in the introduction section on page 4, lines 88 to 96: “To maximize the benefits of strength training, we provided nutrition education on consuming enough protein, to support muscle growth and function (17). However, many older adults do not eat enough protein and consume less than 0.66 grams/kg of body weight/day (19). Food intake in older adults is extremely complex and there are several barriers to consuming enough protein including poor health, poor appetite, changes in food preference, chewing difficulties, etc (20). A population-based study (n = 2,066 community-dwelling adults 70 to 79 years) showed that those consuming at least 1.2 grams of protein/kg of body weight/day lost lean mass over the three-year follow-up period, but this loss was 40% lower compared to those consuming 0.8 grams of protein/kg of body weight/day (21). Initiating exercise when protein intake is inadequate may cause weight loss, or limit capacity or strength gains.”

- Please provide a reference for the statement in line 73-76.

We have included two references to support the development and implementation of balance and functional strength training in older adults – references 10 and 11, page 3, line 76

- Line 78-80 states {Formatting Citation}, please ensure a correct reference.

We have corrected this– references 15 and 16, page 3, line 80. 

- Please provide a reference for the statement in line 87-88.

We have included two citations to support the benefits of protein intake and muscle protein accretion after strength training in older adults – references 17 and 18, page 4, line 88

- The authors state that secondary outcomes were reported in another manuscript in line 96-97, is a reference available?

We will work with the journal to ensure there is a citation before the manuscript is published – we are still waiting to receive feedback on the second manuscript. 

Methods: The study design is reported in great detail and very clear. I have two questions;

- Although I appreciate that the secondary outcomes are reported in a different manuscript, I would advise to briefly describe all the outcome measures. For instance, in line 175 it is stated that the intensity was targeted at <8 repetitions maximum. However, it is unclear when the RM was measured, which exercises etc.

We included the following statement to briefly describe the secondary outcomes: “The secondary outcomes (body weight, gait speed, grip strength, fatigue levels, muscle strength, dynamic balance, health-related quality of life, resource use, and protein intake) were reported in another manuscript (22)” (page 4 and 5 lines 104 to 107).

We did not formally assess 1-repetition maximum, so we have added the following statement to the manuscript: “We did not formally assess 1-repetition maximum. The exercise programs were implemented in real world settings by exercise physiologists, and they were provided a manual and training on how to implement the program. Each exercise physiologist was advised to use informal assessments of multiple repetitions maximum and a repetition in reserve strategy to guide exercise selection and progression. They were instructed to increase the difficulty of the movement if participants could perform more than 8 repetitions” (page 8 to 9, lines 190 to 196). 

- Details for the following might be described elsewhere, but I would advise to briefly describe the control group, as the title and abstract state it is a randomized controlled trial.

This point was also brought up by the first reviewer. Our response was: 

The stepped wedge randomized controlled trial (RCT) design is a type of RCT and is used as an alternative to traditional parallel cluster RCTs . We opted to use a stepped wedge RCT because this type of design provides the advantage that all participants will eventually receive the intervention. When clusters are relatively similar (homogeneous), parallel cluster RCTs tend to deliver better statistical performance than a stepped wedge cluster RCT; however, if there are substantial cluster differences (e.g., population differences such as rural vs urban populations, community-dwelling vs residential), the stepped wedge design will be more powerful than a parallel design . We were considering a stepped wedge RCT for our future pragmatic trial looking at scale-up of our intervention, so we assessed feasibility using the same design we were considering for the larger trial.

The design includes an initial period in which no clusters are exposed to the intervention. Subsequently, at regular intervals one cluster is randomized to cross from the control to the intervention group. In our case, sites three and four act as our “control group” until all groups are allocated to the intervention group. In addition, the CONSORT group developed a separate reporting guideline for stepped-wedged cluster RCTs2. To address your point, we have added the following: “The stepped wedge RCT design is a type of cluster RCT, and is used as an alternative to traditional parallel cluster RCTs”

Results: Clearly described and interesting findings.

Discussion: Well written. The discussion highlights important findings and great insights in regard to the chosen modality of providing the MoveStrong model.

Thank you, we appreciated the comments and feedback.

---

## [Decision Letter · Decision Letter 1]

9 Sep 2021

The MoveStrong program for promoting balance and functional strength training and adequate protein intake in pre-frail older adults: A pilot randomized controlled trial

PONE-D-21-14688R1

Dear Dr. Rodrigues,

We’re pleased to inform you that your manuscript has been judged scientifically suitable for publication and will be formally accepted for publication once it meets all outstanding technical requirements.

Kind regards,

Leonardo A. Peyré-Tartaruga, Ph.D.

Academic Editor

PLOS ONE

Additional Editor Comments (optional):

Reviewers' comments:

Reviewer's Responses to Questions

**Comments to the Author**

1. If the authors have adequately addressed your comments raised in a previous round of review and you feel that this manuscript is now acceptable for publication, you may indicate that here to bypass the “Comments to the Author” section, enter your conflict of interest statement in the “Confidential to Editor” section, and submit your "Accept" recommendation.

Reviewer #1: All comments have been addressed

Reviewer #2: All comments have been addressed

Reviewer #3: All comments have been addressed

Reviewer #4: All comments have been addressed

2. Is the manuscript technically sound, and do the data support the conclusions?

Reviewer #1: Yes

Reviewer #2: Yes

Reviewer #3: Yes

Reviewer #4: Yes

3. Has the statistical analysis been performed appropriately and rigorously? 

Reviewer #1: Yes

Reviewer #2: Yes

Reviewer #3: Yes

Reviewer #4: Yes

4. Have the authors made all data underlying the findings in their manuscript fully available?

Reviewer #1: Yes

Reviewer #2: Yes

Reviewer #3: Yes

Reviewer #4: Yes

5. Is the manuscript presented in an intelligible fashion and written in standard English?

Reviewer #1: Yes

Reviewer #2: Yes

Reviewer #3: Yes

Reviewer #4: Yes

6. Review Comments to the Author

Reviewer #1: I am happy with the authors' responses although I disagree with the statements about feasibility studies not really informing the future main study. First, I never said this is the primary goal of the feasibility study, but the authors ignore it altogether - how will they know the variance of the primary outcome at baseline, for example, if not from this study.

Reviewer #2: (No Response)

Reviewer #3: (No Response)

Reviewer #4: The authors did a thorough job of responding to the comments of all four reviewers. I feel that the authors adequately addressed the comments raised. I have no further requests.

7. PLOS authors have the option to publish the peer review history of their article (what does this mean?). If published, this will include your full peer review and any attached files.

Reviewer #1: No

Reviewer #2: **Yes: **André Ivaniski-Mello

Reviewer #3: **Yes: **Eduardo Lusa Cadore

Reviewer #4: **Yes: **Lara Vlietstra

---

## [Editor Report · Acceptance letter]

15 Sep 2021

PONE-D-21-14688R1 

The MoveStrong program for promoting balance and functional strength training and adequate protein intake in pre-frail older adults: A pilot randomized controlled trial 

Dear Dr. Rodrigues:

I'm pleased to inform you that your manuscript has been deemed suitable for publication in PLOS ONE. Congratulations! Your manuscript is now with our production department. 

Kind regards, 

on behalf of

Professor Leonardo A. Peyré-Tartaruga 

Academic Editor

PLOS ONE